# Beyond Participation: Evaluating the Role of Patients in Designing Oncology Clinical Trials

Eliya Farah [1], Matthew Kenney [1], Anris Kica [1], Paul Haddad [1], David J. Stewart [2] and John-Peter Bradford [1,*]

1    Life-Saving Therapies Network, 173 Heath Street, Ottawa, ON K1H 8L6, Canada
2    Department of Medicine, Faculty of Medicine, The Ottawa Hospital, University of Ottawa, 501 Smyth Rd., Ottawa, ON K1H 8L6, Canada; dstewart@toh.ca
*    Correspondence: jp@lifesavingtherapies.com

**Abstract:** Historically, subject matter experts and healthcare professionals have played a pivotal role in driving oncology clinical trials. Although patients have been key participants, their deliberate and active contribution to the design and decision-making process has been limited. This scoping review aimed to examine the existing literature to scope the extent of active patient engagement in the design of oncology clinical trials and its corresponding influence on trial outcomes. We conducted a systematic search using two databases, namely MEDLINE (Ovid) and EMBASE, to identify relevant studies exploring patient engagement in cancer-related clinical research design. We identified seven studies that met the eligibility criteria. The studies highlighted the benefits of active patient involvement, such as improved recruitment strategies, and the attainment of more patient-centered trial outcomes. The influence of patient involvement varied from tangible developments like patient-friendly resources to indirect impacts like improved patient experiences and potentially higher adherence to trial intervention. The future of clinical trials should prioritize patients' values and perspectives, with regulatory bodies fostering these practices through clear guidelines. As the concept of patient centricity takes root in oncology research, the involvement of patients should evolve beyond mere participation.

**Keywords:** evidence-based patient engagement; oncology clinical trial; patient centricity; design oncology clinical trial

## 1. Introduction

Clinical trials have historically been a pivotal platform for innovations in cancer care and control, profoundly influencing the medical landscape over time [1]. Their findings accelerate the discovery and validation of novel interventions, thereby directly influencing the health outcomes and quality of life of countless patients worldwide [2]. Well-conducted trials are regarded as the gold standard for producing reliable clinical evidence across various medical specialties, including oncology [3]. Historically, pharmaceutical companies have played a major role in shaping the design and framework of these trials. As primary funders, they may engage healthcare professionals and subject matter experts who align with the interests and priorities of the pharmaceutical company [4,5]. This influence might be evident in various aspects of the trial design, particularly in the strategic selection of the eligibility criteria (i.e., exclusion and inclusion criteria). These criteria may be designed in a manner that selectively omits certain patient populations, potentially leading to bias in favor of participants with more favorable prognoses [6]. Traditionally, patients, have been relegated to participants, receiving treatments and contributing data but having little influence over the trial's design or execution [7]. This approach, while expert-driven, often overlooked the values and preferences of patients and/or skewed the results in favor of the treatments, leaving an important aspect of healthcare under-represented [8].

The growing emphasis on patient-centered care has highlighted the importance of integrating patient values and perspectives into cancer management and research development [9,10]. Firsthand experiences of patients may offer valuable insights into disease management as well as the delivery of interventions and adherence. Moreover, the pharmaceutical industry is witnessing a growing trend toward patient centricity, where patient needs inform strategic decisions across the continuum of care. This shift in perspective recognizes patients as active contributors to the research process, going beyond mere participation [11]. Today, patients are being incrementally involved in decision making, trial design, management, organization, and result interpretation [12]. This patient-centric approach acknowledges that patients bring unique insights to research designs and trial execution [12,13]. By incorporating patient values and preferences, clinical trials can achieve improved recruitment and retention rates, and generate high data quality [14].

In light of recent advancements in oncology research, there is a discernible shift toward incorporating patient-centric approaches, especially with the increased emphasis on personalized medicine and tailored therapies [15–17]. This patient-centered approach is not merely an academic preference but is increasingly being endorsed by regulatory bodies. For instance, the Canadian Agency for Drugs and Technologies in Health (CADTH), the National Institutes of Health (NIH) in the United States, and the European Medicines Agency (EMA) in Europe have issued guidelines and a call to action to integrate patient perspectives and values into clinical research, including but not limited to, study design, execution, and interpretation [14,18,19]. Moreover, agencies such as the Patient-Centered Outcomes Research Institute (PCORI) have provided detailed methodologies and frameworks for involving patients in various stages of clinical research, thereby underscoring the institutional support for this paradigm shift [20]. These pivotal developments indicate that the inclusion of active patient involvement is not an optional addition but an evolving standard in oncology research.

Despite recognizing the value of patient participation in oncology trials, there remains a gap in their active involvement in trial design [21]. This scoping review aims to investigate the existing literature on active patient involvement in the design of oncology clinical trials and examine its consequent impact on trial outcomes. In this review, we will identify and consolidate studies that have explored patient involvement in various stages of trial design, including protocol development, decision-making processes, and result interpretation. As we synthesize the available evidence, we seek to provide a comprehensive overview of the current landscape, highlighting the potential benefits and challenges associated with such involvement. The findings of this review will contribute to a better understanding of the value of active patient engagement in optimizing the design and outcomes of oncology clinical trials and how this can inform future research and practice in this developing area.

## 2. Materials and Methods

### 2.1. Search Strategy and Selection Criteria

In compliance with the guidelines presented by Munn et al., as well as the Preferred Reporting Items for Systematic Reviews and Meta-Analyses (PRISMA), we chose to conduct a scoping review as opposed to a systematic review [22,23]. Also, we opted for a scoping review as opposed to a systematic review for several key reasons. A scoping review allows for the inclusion of a broad range of study designs and methodologies, thereby offering a more comprehensive overview of the existing literature. This approach is particularly suited to our research question, which aims to investigate a multifaceted and evolving topic: the active evidence-based involvement of patients in the design of oncology clinical trials.

We systematically searched two electronic databases, namely MEDLINE (Ovid) and EMBASE, to identify relevant studies on the intersection of oncology, patient involvement, and clinical trial research. In our search strategy, we used a combination of index keywords, MeSH, and iterative search terms, aiming to capture studies that explored patient engagement in cancer-related clinical research design (Supplementary Table S1). We applied no

language restrictions to ensure the inclusion of relevant studies published in any language. The search was last conducted in adherence to PRISMA guidelines on 12 December 2022.

*2.2. Eligibility Assessment*

Our goal was to encompass a wide range of articles that demonstrate the various ways in which patients are involved in the design of oncology clinical trials. The inclusion criteria for this study included research articles that met the following criteria: (1) proposed programs and methods for patient involvement, and integrated patient input and feedback in the design, methodology, and/or treatment choices; (2) utilized patient-reported outcomes in trial design; and (3) the clinical trial must have been oncology-related, involving patients diagnosed with cancer of any type. Additionally, the included papers were required to (4) measure outcomes specifically related to patient involvement.

We excluded studies from our analysis that merely provided patient-reported outcome measures without actively integrating them into the trial's design and execution. We also dismissed gray literature, correspondences, posters, literature reviews, communications, and other non-primary research sources.

We utilized EndNote X9 [24] reference management software to import all records and remove any duplicates. The remaining records were transferred to the COVIDENCE [25] web platform, where additional duplicate records were filtered out. In the initial round, two reviewers (E.F. and A.K.) independently screened the titles and abstracts of the captured records. Reviewers then reached a consensus on discrepancies in decisions. Subsequently, full-text eligibility screening was performed and independently validated by the same two reviewers (E.F. and A.K.). We invited a third (P.H.) and a fourth (M.K.) reviewer who independently reviewed and validated the full-text screening results.

*2.3. Data Abstraction*

Data from all included records were independently abstracted by three reviewers (E.F., M.K., and P.H.). One reviewer (E.F.) then verified all abstracted data. For each article included, we extracted the following information: the first author's name, the year of publication, the design of the study, the country in which the research was carried out, the cancer type investigated, the stage of cancer, the status of the patient as a stakeholder (active, survivor, or caregiver), the number of patients involved, the number of other stakeholders (i.e., clinicians, researchers, general public, etc.), the point at which patients were engaged in the research process (i.e., planning phrase, study design, etc.), whether there was training or prior experience to patients being recruited, the source of patient recruitment, and the strategy taken to recruitment (Table 1).

As shown in Table 2, we focused on extracting the following variables: the method used for patient engagement (i.e., web conferences, in-person meetings, focus groups, etc.), how often patients were engaged, whether patients were compensated for their participation, any reported advantages arising from patient involvement, the specific manners in which patient feedback shaped the trial design, any identified challenges or hurdles related to patient participation, the trial outcomes and any possible influence from patient involvement on these, and if and how the impact of patient engagement was assessed.

## 3. Results

*3.1. Identified Studies*

Figure 1 illustrates the search results for relevant studies and the screening process. Of the 2768 records identified, 482 duplicates were removed. Based on title and abstract, 2215 were excluded, leaving 71 full-text articles to be retrieved and assessed for eligibility. Of these, 64 were excluded for reasons outlined in Figure 1. A total of seven articles were included in our scoping review.

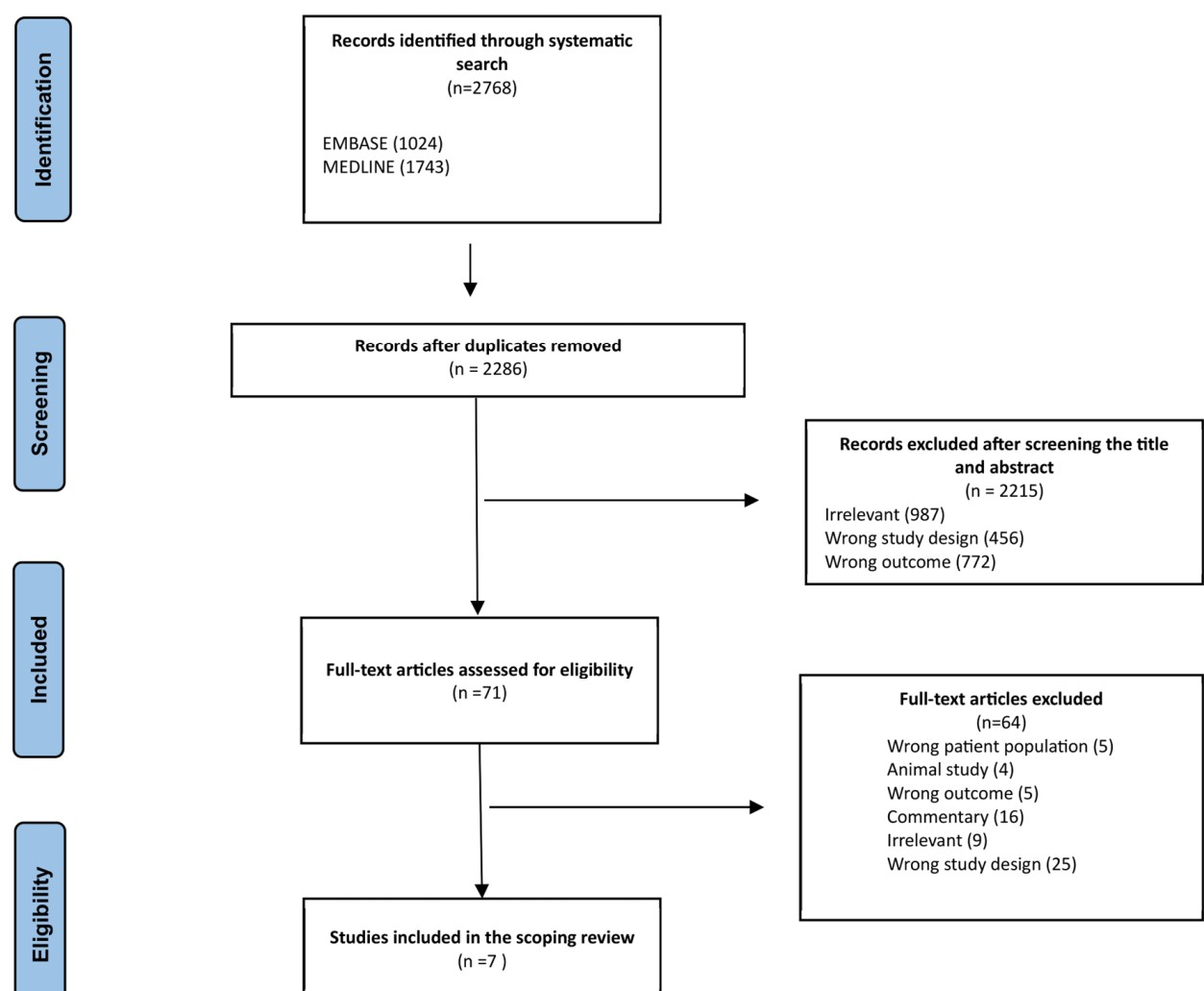

**Figure 1.** Preferred Reporting Items for Systematic Reviews and Meta-Analyses (PRISMA) flow diagram for the literature search and study selection. After a systematic search using two databases, namely MEDLINE (Ovid) and EMBASE, seven studies were identified that met the eligibility criteria.

*3.2. Study Characteristics*

The data, as summarized in Table 1, exhibit a range of pragmatic clinical trials conducted between 2004 and 2022, located primarily in the United States (three studies) and the United Kingdom (three studies), with one study in Denmark [26–32]. The trials involved patients with different cancer types, including breast (three studies), esophageal (one study), and high-grade non-muscle-invasive bladder cancer (one study); one study included multiple cancer types, namely breast, colorectal, and non-small cell lung [26–32]. The patient populations in these trials consisted of individuals at different cancer stages, including early stages (two studies), advanced stages (two studies), and mixed stages (one study), as well as cases where the stage was not specified (two studies) [26–32]. The study designs employed included pragmatic clinical trials (six studies) and a pragmatic observational trial (one study).

Across the studies, the number of patients involved in shaping the trial design ranged from as few as 7 to as many as 286, reflecting a wide array of patient involvement in clinical design. Similarly, the extent of involvement from other stakeholders varied, ranging from 7 to 54 participants. Regarding patient engagement in the research process, the study design phase emerged as the most prevalent point of involvement, with three studies emphasizing this phase. Additionally, two studies extended patient engagement to the planning phase,

one study to interpretation, and one study to dissemination of results. Regarding the training provided or previous experience, one study offered training to patients.

Patient recruitment sources were diverse, with studies utilizing patient advocacy groups (one study), hospitals (three studies), cancer charities (one study), social media and existing patient networks (one study), and a combination of university hospital departments and treating clinicians (one study) [26–32]. The recruitment strategies employed were equally varied, including website advertisements, flyers, and in-person approaches (one study); hospital record screening followed by in-person approaches (one study); invitations through newsletters and telephone (one study); advertisements via social media and direct personal invitations (one study); and identification by site principal investigators or organizational leadership (one study); two studies did not specify their recruitment strategies [26,31].

Table 2 provides an overview of the assessment of patient engagement in clinical trials, highlighting various engagement methods, impacts, challenges, outcomes, and the evaluation of effectiveness across the seven studies included [27–32]. The studies encompassed a range of engagement methods, with one study employing web conferences, patient partner-specific web conferences, in-person meetings, and email correspondence. Another study utilized focus groups and in-depth interviews, one engaged participants through focus group meetings and individual interviews, and one conducted focus groups and meetings. Two studies facilitated focus group sessions with presentations and employed semi-structured open-ended questions, while one utilized virtual meetings, surveys, discussions, and scientific presentations. The final study conducted 1:1 structured interviews, workshops, 1:1 feedback sessions, and stakeholder meetings.

The frequency of engagement also varied among the studies. One study engaged participants yearly over a 4-year period. Another conducted pre- and post-focus group interviews over 6 months, and one held nine focus groups and two meetings over a specific duration. One study conducted three focus group sessions over a 7-month period, while another facilitated collaboration over a 2-year period, without specifying the exact frequency. One study held four stakeholder meetings in the first 18 months.

**Table 1.** Overview of the 7 studies examining patient involvement in oncology-related clinical trials: variations across cancer types, stages, and recruitment strategies.

| Authors | Study Design | Country | Cancer Type | Stage | Status of Patient Stakeholder | Number of Patients | Number of Other Stakeholders | Stage of Research in Which Patients Were Engaged | Training Provided or Previous Experience | Recruitment Source | Recruitment Strategy |
|---|---|---|---|---|---|---|---|---|---|---|---|
| Barger et al., 2019 [26] | Pragmatic Clinical Trial | United States | Breast, Colorectal, or Non-Small Cell Lung | 0, I, IA, IB, IIA, IIB, IIC, IIIA, IIIB, IIIC, IV, IVA, IVB | Active Patient, Survivor, Caregiver | 10 | 11 | Planning phase, study design, interpretation, and dissemination of results. | Yes | 6 from the SWOG Patient Advocate Committee | NA |
| Forbes et al., 2010 [28] | Pragmatic Clinical Trial | UK | Breast | NA | Survivor, Other | 15 | 54 | Study design | No | Cancer charities: (1) Macmillan (2) Breast Cancer Care (3) Asian Women's Breast Cancer Support Group | (1) Website (2) Flyers and newsletter (3) Approached women in public areas |
| Hoeg et al., 2019 [27] | Pragmatic Clinical Trial | Denmark | Breast | NA | Survivor | 7 | NA | Study design | No | New Zealand University Hospital, Department of Oncology | Hospital record screening by two nurses followed by approaching patients in person |
| Marsden et al., 2004 [29] | Pragmatic Clinical Trial | UK | Breast | I and II | Active Patient, Survivor | 83 | 7 | Study design | No | (1) Consumers' Advisory Group for Clinical Trials (CAG-CT) (2) Participants of the pilot HRT study (3) The Lynda Jackson Macmillan Centre at Mount Vernon Hospital (4) CAG-CT (5) Clinicians involved in the undertaking of the previously described pilot HRT study | (1) Invitations to participate via newsletters (2) Telephone |
| Nicholas et al., 2021 [30] | Pragmatic Clinical Trial | UK | Esophageal | cT1-4a and/or cN+, cM0 | Active Patient, Survivor, and Caregiver | 21 | NA | Study design | No | (1) Manchester University NHS Trust patient engagement teams (2) Wales Cancer Research Centre (3) Treating clinicians who identified potential patients (4) Existing patient networks, such as esophageal cancer support groups and other cancer support centers | (1) Sent out adverts via social media networks to regular patient contributors. (2) Disseminated adverts through social media networks and regular patient contributors. (3) Sent personal invitations directly to individual patients. (4) Advertised meetings within these support groups and centers. |

**Table 1.** *Cont.*

| Authors | Study Design | Country | Cancer Type | Stage | Status of Patient Stakeholder | Number of Patients | Number of Other Stakeholders | Stage of Research in Which Patients Were Engaged | Training Provided or Previous Experience | Recruitment Source | Recruitment Strategy |
|---|---|---|---|---|---|---|---|---|---|---|---|
| Smith et al., 2022 [31] | Pragmatic Observational Trial | United States | High-grade Non-Muscle-Invasive Bladder Cancer | Ta, T1, CIS, T2, T3, T4 | Active Patient, Caregiver | 286 | NA | The engagement plan guided PPI throughout the stages of the study, including study design, conduct, analysis, and dissemination. | No | BCAN PSN (Bladder Cancer Patient Survey Network) | Not Specified |
| Solomon et al., 2017 [32] | Pragmatic Clinical Trial | United States | Lung, Head and Neck, Sarcoma, Prostate, Ovarian, Colorectal, Melanoma, Glioblastoma | Advanced Cancer | Active Patient, and Caregiver | 12 | 15 | Study design | No | Four study centers (an academic, a municipal, and a community hospital in NYC and a rural hospital in Connecticut) | Identified by site principal investigators (PIs) or their organizational leadership |

*3.3. Challenges to Patient Engagement and Reported Benefits*

Studies reported several challenges and barriers to patient involvement. Among these, the significant time and financial commitments required for focus groups, meetings, and interview execution emerge as a substantial obstacle, potentially stretching the resource capacities of the research team. The incongruity between lay and expert approaches also presents a unique challenge, as the task of reconciling patient insights with the technical and scientific requirements of research can demand considerable effort, sensitivity, and time. In addition, the need to align patient experiences with the strict protocols and guidelines established with the hierarchy of scientific evidence adds another layer of complexity, requiring a deft integration of qualitative patient input into a largely quantitative empirical framework. Furthermore, achieving a representative patient sample—crucial for the generalizability of findings—often proves to be difficult due to variability in demographics and disease stages, as well as potential access and availability barriers, particularly among under-represented or marginalized groups (i.e., rural, LGTBQ2S+, immigrants, minor ethnic groups, etc.) (Table 2).

The reported benefits of patient involvement were diverse across the board. Certain studies reported direct outcomes, such as the development of patient-friendly materials and intervention modifications, while others noted possible indirect impacts like improved patient experience and potentially higher participation rates. For example, in Barger et al. (2019), we observed the long-term benefits of strategic patient engagement. Communication via web conferences and emails improved the trial's design and implementation [26]. In addition, patient input led to notable enhancements such as refining study endpoints and aiding in decision-making processes [26]. From Forbes et al. (2010), we observed the value of patient opinions on the consent process [28]. The authors reported that patients were more comfortable with an opt-out consent approach and preferred verbal over written consent for interventions [28]. In the study by Hoeg et al. (2019), patient feedback directly influenced several aspects of the research, such as recruitment strategy and the creation of educational materials [27]. Marsden et al. (2004) highlighted how patient engagement could lead to the recognition of significant trial endpoints, such as quality of life, that researchers might otherwise overlook [29]. Also, patient input improved the information flow regarding side effects and trial updates [29]. In Nicholas et al. (2021), patient feedback led to equitable access for participants, improved trial feasibility, and a focus on patient-preferred outcomes such as toxicity reduction [30]. In Smith et al. (2022), patients actively participated in creating user-friendly trial materials [31]. They also influenced the decision to shift toward a more observational study type. Lastly, in the study by Solomon et al. (2017), patient involvement led to an intervention better suited to the needs of physicians and patients, therefore refining the efficacy of the trial [31] (Table 2).

**Table 2.** Assessment of patient engagement in clinical trials: engagement methods, impact, challenges, and outcomes.

| Authors | Engagement Method (Frequency) | Engagement Frequency | Benefits of Patient Involvement | Specific Ways in Which Patient Input Influenced the Trial Design | Reported Challenges or Barriers to Patient Involvement | Outcomes of the Trial Influenced by Patient Involvement | Whether and How the Effectiveness of Patient Engagement Was Evaluated | Compensation for Patients |
|---|---|---|---|---|---|---|---|---|
| Barger et al., 2019 [26] | • Web conferences<br>• patient partner-specific web conferences<br>• In-person meetings<br>• Email correspondence | • Yearly over a 4-year period: two web conferences, two patient partner-specific web conferences, one in-person meeting | • Critical thinking,<br>• Unique insights<br>• Collaborative problem-solving.<br>• Improved trial design and implementation | • Revised study endpoints<br>• Guidance on FN risk algorithm<br>• Reviewed regimens and FN risk levels<br>• Advised on cohort and usual care arms<br>• Included additional questions in patient surveys<br>• Recommended incorporating pharmacy-related questions<br>• Provided feedback on study statements<br>• Advised on lay language, descriptive diagram, financial resources, recruitment and results dissemination strategies | • NA | • Created best practices for stakeholder engagement in healthcare research and clinical trials | • Annual stakeholder satisfaction, engagement, and impact survey | • Yes |
| Forbes et al., 2010 [28] | • Focus groups<br>• In-depth interviews | • Seven focus groups, seventeen in-depth interviews | • Insights leading to protocol and information improvements | • Support for opt-out consent,<br>• Preference for verbal consent<br>• Emphasis on informed consent<br>• Acceptance of routine datasets·Coordination of contact with breast cancer patients by the breast care team | • Expenses: focus groups and in-depth interviews requiring trained researchers and significant effort to develop materials<br>• Time commitment | • Acceptance of opt-out consent for low-risk trials<br>• Need for robust data governance and participant trust<br>• Accommodating participant preferences<br>• Acceptance of routine datasets with strong governance | • NA | • Yes |

**Table 2.** *Cont.*

| Authors | Engagement Method (Frequency) | Engagement Frequency | Benefits of Patient Involvement | Specific Ways in Which Patient Input Influenced the Trial Design | Reported Challenges or Barriers to Patient Involvement | Outcomes of the Trial Influenced by Patient Involvement | Whether and How the Effectiveness of Patient Engagement Was Evaluated | Compensation for Patients |
|---|---|---|---|---|---|---|---|---|
| Hoeg et al., 2019 [27] | • Focus group meeting<br>• Individual interviews (both pre- and post-focus group interviews) | • Pre- and post-focus group interviews conducted 2 weeks before and after the focus group, all interviews conducted over 6 months | • Improvements in recruitment strategies, user-friendly information, strong support from nurses, and approached patients | • Changes in recruitment strategy, brochures, and educational material<br>• Creation of an electronic platform for questionnaire data collection<br>• Refinement of questionnaire items<br>• Addressing patient concerns<br>• Reconciling patient perspectives with research criteria and evidence hierarchy<br>• Prioritizing the patient's voice<br>• Developing new outcome measurement hierarchies | • Challenges in describing panel member roles<br>• Confusion among patients about trial participation<br>• Conflicting perspectives between lay and expert approaches<br>• Difficulties in reconciling patient perspectives with research criteria and evidence hierarchy<br>• Representativeness of patient population | • NA | • In the post-focus group interview, panel members were asked to assess the involvement process using an evaluation form that included a visual representation of the cube model.<br>• Nurses were interviewed specifically about the screening and recruitment process. | • No |
| Marsden et al., 2004 [29] | • Focus groups<br>• Meetings | • Nine Groups (6 groups of women from breast cancer support groups: 3 included women participating in the pilot HRT study)<br>• Two Meetings (Meeting 1—duration of half a day, Meeting 2—duration of 1 day) | • Identification of issues relevant to breast cancer patients<br>• Formulation of practical priorities for trial design | • Emphasis on quality of life as an endpoint, preference for no placebo arm<br>• Suggestions on the timing of patient invitations | • Negotiating conflicting goals between patients and clinicians<br>• Ensuring adequate information for informed consent and decision making | • Production of a patient booklet<br>• Improved provision of information about side effects<br>• Strategies for trial updates | • NA | • Not stated |

**Table 2.** *Cont.*

| Authors | Engagement Method (Frequency) | Engagement Frequency | Benefits of Patient Involvement | Specific Ways in Which Patient Input Influenced the Trial Design | Reported Challenges or Barriers to Patient Involvement | Outcomes of the Trial Influenced by Patient Involvement | Whether and How the Effectiveness of Patient Engagement Was Evaluated | Compensation for Patients |
|---|---|---|---|---|---|---|---|---|
| Nicholas et al., 2021 [30] | • Facilitated focus group sessions with presentations (investigators)<br>• Semi-structured open-ended questions (investigators and facilitators) | • Three focus groups, over a 7-month period (November 2018 to June 2019). | • Embedding patients views in trial design<br>• Promoting patient-centered outcomes<br>• Addressing financial barriers<br>• Providing insights on treatment preferences | • Support provision<br>• Travel expenses coverage<br>• Randomization approach<br>• Fraction schedule<br>• Trial endpoints<br>• Patient information materials<br>• Communication protocols | • Ensuring diversity and representation<br>• Addressing inequality in access to Proton Beam Therapy | • Not specified, but a patient-centered approach is expected to improve enrolment, feasibility, and impact | • The document does not provide specific details about the evaluation process but suggests that patient involvement had a tangible impact on trial design. | • Yes |
| Smith et al., 2022 [31] | • Virtual meetings<br>• Surveys<br>• Discussions<br>• Scientific presentations. | • Collaboration facilitated over a 2-year period (frequency not specified) | • NA | • Improved patient-facing materials<br>• larified survey language<br>• Considerations for emotional well-being<br>• Preference for observational trial over RCT | • Disagreements among diverse groups<br>• Potential misinterpretation of clinical terminology<br>• Initial engagement difficulties | • Not specified, but patient feedback is expected to enhance patient experience and potentially improve response rates, indirectly influencing trial outcomes | • An engagement evaluation was conducted, with high ratings in some areas and areas identified for improvement. | • Not stated |
| Solomon et al., 2017 [32] | • 1:1 Structured interviews<br>• Workshops<br>• 1:1 Feedback sessions<br>• Stakeholder meetings | • Four stakeholder meetings in the first 18 months | • Tailored intervention for physicians and patients<br>• Improved measurement of intervention impacts | • Changes in proposal development<br>• Intervention modification·<br>Outcome measure refinement<br>• Clarification of goals-of-care discussions<br>• Exploration of underlying concepts | • Ensuring participation among a sick patient cohort<br>• Difficulties in finding interested and physically able participants<br>• Reproducibility of physician participation<br>• Unique contextual factors<br>• Communication skills | • High participation rates among oncologists<br>• Development of a novel communication training model (joint visit) | • NA | • Yes |

*3.4. Patient-Centric Trial Framework*

Figure 2 presents a detailed framework highlighting the significance of meaningful patient engagement throughout the multiple stages of oncology clinical trials. In this proposed structure, we simplified the trial process into six key stages: the initial trial design and development, participant recruitment and consent, trial implementation and monitoring, data management and analysis, the dissemination of results and post-trial activities, and future research and trial replication.

In each stage, we identified specific opportunities for patients to share their insights and perspectives, thereby improving patient engagement in the trial. This is important because patients can enrich the trial design by helping define more pertinent research questions, shaping more patient-friendly recruitment and consent processes, and guiding the interpretation and dissemination of trial results in a manner that is easily understood and beneficial for the patient community.

In the initial trial design and development phase, patients collaborate to define essential research questions that resonate with their experiences. Their unique insights illuminate unexplored aspects of the study, enriching trial objectives, hypotheses, and endpoints. Patient engagement extends into participant recruitment and consent, where patients' firsthand knowledge helps identify potential barriers to participation. By providing practical strategies to overcome these obstacles, patients ensure successful enrollment. Furthermore, their guidance on informed consent materials aids in crafting clear, patient-friendly explanations of trial details.

As trials transition to the implementation and monitoring phase, patients offer practical guidance on logistical considerations, optimizing trial execution in real-world settings. Their involvement in monitoring committees adds a safety dimension, affirming the trial's integrity. During data management and analysis, patients advocate for capturing holistic health aspects and refining the measurement of relevant outcomes. Their input also shapes the analysis plan and result interpretation, yielding insights with direct patient relevance.

The dissemination of trial results involves patients in translating statistical findings into clinically meaningful insights, enhancing accessibility by conveying outcomes in patient-friendly language. Their active role in monitoring late-emerging effects and contributing to trial closure underscores their commitment beyond the trial's formal conclusion. In envisioning future research and trial replication, patients' experiences and feedback directly inform the planning of subsequent studies, ensuring that new endeavors address patient priorities.

The framework depicted in Figure 2 underscores the value of moving toward a patient-centric approach in oncology clinical trials. By integrating patient perspectives at every step, we can ensure that the outcomes of the trials align with the needs and priorities of the very individuals they are intended to benefit.

Figure 2 offers a comprehensive framework for patient involvement in the design and execution of oncology clinical trials. This figure highlights the various stages of a clinical trial, from the initial trial design to future research and trial replication. It further details the key activities associated with each stage. The central aspect of the figure illustrates the potential points of patient engagement at each stage, emphasizing their potential contributions, including insightful input, feedback, implementation guidance, and more. This framework underpins the multifaceted role of patients in enhancing the relevance, conduct, and impact of oncology clinical trials.

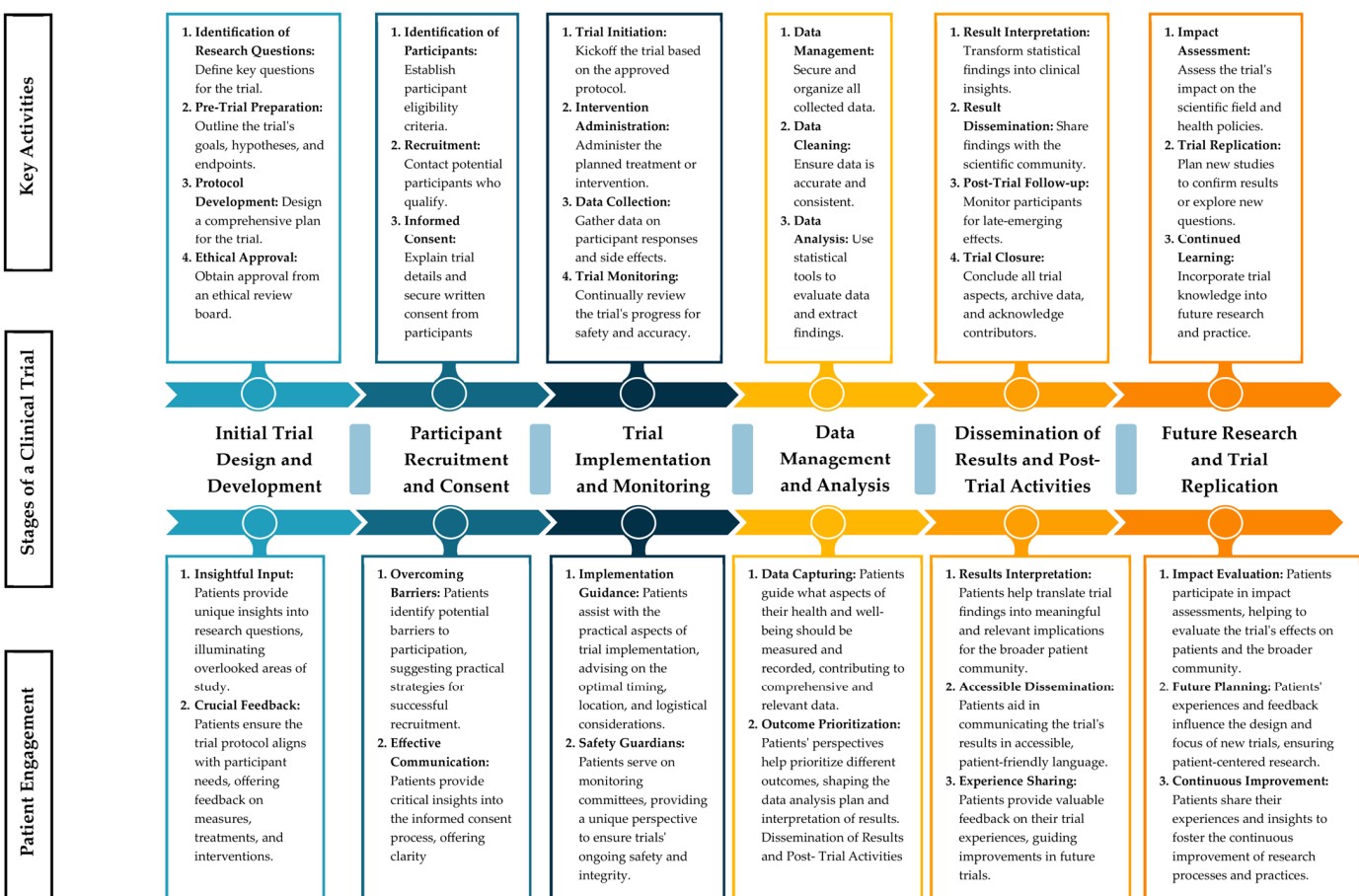

**Figure 2.** A comprehensive framework for patient involvement in the design and execution of oncology clinical trials.

## 4. Discussion

We conducted a scoping review to explore the role of active patient involvement in the design of oncology clinical trials and evaluate its influence on trial outcomes. Our consolidation of the available literature highlighted a potential for patient involvement in shaping trial design, influencing a range of aspects from protocol development to result interpretation. These benefits include the enrichment of trial endpoints, the optimization of recruitment strategies, the enhancement of patient-centric outcomes, and the facilitation of more understandable, user-friendly interpretation of trial-related information.

Patients can play an instrumental, partnership role in various stages of a clinical trial, starting from the initial design and development, where their unique insights and experiences could guide the formation of research questions and shape a study's framework [32,33]. Their input becomes crucial during pre-trial preparations, with feedback on measures, treatments, and interventions contributing to a protocol that is reflective of and responsive to the needs and preferences of participants [34]. During the recruitment and consent phase, patients can identify potential barriers to participation and offer strategies for their mitigation, therefore improving overall recruitment efforts [35]. Their insights are equally critical in facilitating informed consent through clear, patient-oriented communication of necessary information and legal documents [36]. As trials move toward implementation and monitoring, patients continue to make important contributions by advising on practical aspects such as timing, location, and logistics [36,37]. Their involvement in data monitoring committees helps safeguard participants and promptly address emergent issues [38]. Patients' involvement in data management and analysis, mainly through participation in data monitoring committees, helps safeguard participants, address emer-

gent issues promptly, and contribute valuable insights for enhanced data oversight [38,39]. Following the completion of the trial, the collaboration of patients in interpreting and disseminating results ensures the findings are presented in a patient-relevant manner and facilitates effective communication through accessible channels. During post-trial activities, patient feedback becomes a vital tool for refining future trials [40]. Their participation in impact assessments offers a lens to understand the trial's effect on patients and the community. These contributions could impact how trials are designed in the future, accelerating the shift from traditional research methods to ones that prioritize the needs of patients.

Regulatory organizations such as the Food and Drug Administration (FDA) and CADTH have recognized the trend toward patient-centric healthcare and are increasingly advocating for patient involvement in clinical trials [41]. These organizations recognize the important role of patients in enhancing trial design, streamlining regulatory oversight, and ultimately promoting improved treatment outcomes. This pivot toward a more patient-centered model signifies a broader trend in healthcare that acknowledges patients as key stakeholders [42]. Beyond mere recognition, these regulatory institutions have a crucial role in promoting and standardizing patient engagement in clinical trials. The development and implementation of comprehensive guidelines by these organizations can foster the widespread adoption of patient-centered practices. These guidelines should emphasize that patient engagement extends beyond a token gesture, instead serving as a meaningful, evidence-based approach that influences every step from trial design to execution. Therefore, it is important to underscore that the simple inclusion of patients in the process is insufficient.

Patient involvement in clinical trials encounters several significant challenges, including logistical issues such as the time and resources required to facilitate in-depth patient engagement, alongside a scientific discord between medical experts and lay perspectives on research [43,44]. Other challenges involve integrating patient insights into the traditional hierarchical structure of scientific evidence while ensuring a diverse and representative patient sample [43–46]. Various strategies can be used to address these challenges. These include attempts to empower patients through patient education, the creation of standardized frameworks for patient involvement, and dedicated efforts to ensure diversity in patient representation across demographic factors. In the same way that pharmaceutical companies may have biases that influence clinical trial designs toward more favorable outcomes, some patient groups might have preconceived notions that could adversely impact the course of clinical research. To mitigate these challenges, we propose a partnership model in which patients, acting as informed participants, co-lead trial development and execution [47,48]. It is essential that patients bring not only their experiences and values but also a commitment to evidence-based outcomes, and research literacy [49]. As such, patients with requisite skills and experiences should be integral in research development and healthcare technology assessment [49,50].

Patient involvement in clinical trials encounters several significant challenges, including logistical issues such as the time and resources required to facilitate in-depth patient engagement, alongside a scientific discord between medical experts and lay perspectives on research. To directly address these challenges, we propose four practical pathways: (1) making it mandatory to involve patients in the design of clinical trials and any health technology assessment (HTA) processes; (2) implementing a "Patient Advisory Board" at the planning stage to guide trial design and ensure patient-centered endpoints; (3) utilizing digital platforms for virtual consultations to reduce the logistical burden on patients, thereby making it easier for them to participate; and (4) instituting diversity quotas to ensure that the patient sample is representative of different demographic and clinical populations. Importantly, mere patient involvement is not sufficient; it is essential that participating patients bring not only their lived experiences but also a commitment to evidence-based outcomes, bolstered by a level of scientific literacy and research understanding. By implementing these targeted strategies, we can enhance both the quality and applicability of our research, fostering a more patient-centric approach to clinical trials.

Patient participation can be of potential value beyond just the design and execution of the trial. Once an effective drug is approved, patients need to have access to it. The regulatory system in Canada, including CADTH processes and the Patented Medicine Prices Review Board (PMPRB), often undermines and/or marginalizes patient meaningful input in regulations surrounding drug access, which often lack differentiation between drugs intended for severely ill patients and those used for minor ailments [51]. This blanket approach does not take into account the very different risk–reward decision-making criteria facing patients with lethal diseases, who are virtually certain of near-term death and poor quality of life without access to promising treatments. The current approach not only impedes timely access to promising treatments for patients with life-threatening conditions but also makes them more costly [52]. These delays and costs lead to a significant loss of life years [53,54]. Patients can significantly contribute to clinical research reform in several ways [55]. They can help reshape access to trials by revising eligibility criteria. They can suggest accelerated paths to promising treatments, such as enhancing special access programs or approving therapies based on early-phase results, thereby circumventing time-intensive phase III trials. They can also advise on decentralized efficacy tracking in real-world settings [55]. Patients may further support the development of flexible methodologies that allow, for example, protocol deviations without official amendments and broader, simpler access to the trial. Lastly, their involvement can simplify embedded bureaucratic procedures in trial protocols, and more [55]. By overcoming these challenges, we can strive for more than just patient engagement; we can cultivate an informative and diverse patient contribution that leads to an evidence-based, patient-centric approach, thereby elevating the relevance and impact of clinical trials.

While our scoping review has contributed valuable insights into the role of patient involvement in oncology clinical trials, it is important to acknowledge inherent limitations. Our focus on published articles may have excluded gray literature that could offer additional perspectives. Furthermore, the scoping review methodology limits our analysis to exploratory synthesis, thus lacking the robust risk-of-bias assessments that are characteristic of systematic reviews. However, it is worth noting that the scoping methodology allowed for a broader, more inclusive assessment of diverse research activities and patient engagement strategies, serving as a comprehensive starting point for future in-depth systematic reviews. This approach enabled us to capture a wider spectrum of studies, thereby providing a more holistic overview of the landscape of patient involvement in oncology trials.

## 5. Conclusions

This scoping review underscores the crucial role of patient involvement in oncology clinical trials, signifying a paradigm shift in cancer research toward a patient-centric model. By valuing patients as partners and incorporating their unique perspectives, we can enhance recruitment, retention, and data quality while ensuring that the trial outcomes align with patient priorities. Although challenges persist, such as reconciling expert and lay perspectives and ensuring diverse patient representation, these can be overcome with targeted strategies, including patient education initiatives and structured frameworks for patient involvement. Regulatory bodies also hold an essential role in fostering this patient-centric model through the provision of clear guidelines. As we move forward, patient engagement should not be considered a mere procedural step but a valuable source of insight, influencing every aspect of clinical trial design and execution. This shift has the potential to not only redefine clinical trials but also to transform the overall approach to oncology clinical trials. Considering our findings, future research could focus on the development of standardized guidelines for patient involvement in oncology clinical trials, particularly emphasizing evidence-based contributions from educated and scientifically literate patients. Investigating the role of patient education programs in preparing patients for meaningful engagement could also be a pivotal next step, offering new ways to reconcile expert and lay perspectives effectively.

**Supplementary Materials:** The following supporting information can be downloaded at: https://www.mdpi.com/article/10.3390/curroncol30090603/s1, Table S1: Search strategy used to identify relevant studies on the active engagement of patients in the design of oncology clinical trials.

**Author Contributions:** Conceptualization, A.K. and J.-P.B.; methodology, E.F., M.K., P.H., A.K., D.J.S. and J.-P.B.; validation A.K., D.J.S. and J.-P.B.; formal analysis, E.F., M.K. and P.H.; data curation, E.F., M.K. and P.H.; writing—original draft preparation, E.F., M.K. and P.H.; writing—review and editing, E.F., M.K., P.H., A.K., D.J.S. and J.-P.B.; visualization, E.F., M.K. and P.H.; supervision, A.K., D.J.S. and J.-P.B.; project administration, A.K. All authors have read and agreed to the published version of the manuscript.

**Funding:** This research received no external funding.

**Conflicts of Interest:** The authors declare no conflict of interest.

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
