# Peer review of "Beyond Participation: Evaluating the Role of Patients in Designing Oncology Clinical Trials"

_curroncol, doi:10.3390/curroncol30090603_

Round 1
Reviewer 1 Report
I would like to thank the handling editor for offering me the opportunity to review the manuscript entitled “Beyond participation: evaluating the role of patients in designing oncology clinical trials” authored by Farah and colleagues, which is currently under consideration for publication in Current Oncology. I would also like to commend the authors for their scholarly work, which presents a scoping review that investigates the existing literature on the role of active patient engagement in the design of oncology clinical trials. The authors conducted a systematic search of two databases, MEDLINE and EMBASE, to identify relevant studies exploring patient involvement in cancer-related clinical research design. Their search identified seven studies that met the eligibility criteria for inclusion in the review. These studies examined patient engagement across various cancer types and stages, utilizing methods like focus groups, interviews, and workshops to obtain patient input.
The review highlights that patient involvement led to tangible impacts on trial design, such as refinements to recruitment strategies, study endpoints, and patient-facing materials. Patients also provided insights that emphasized patient-cantered outcomes, such as quality of life and toxicity reduction. However, the authors note that patient engagement encountered challenges as well, including resource constraints, reconciling lay and expert perspectives, and ensuring representative patient samples.
Overall, the review underscores the value of active patient involvement in optimizing oncology trial design and outcomes. The authors conclude that patients can play a partnership role across all stages of clinical trials, from shaping the research questions to interpreting the results. They advocate for regulatory bodies to promote patient-centric practices through guidelines and standards. The review provides a timely overview of the evolving role of patients as collaborators and contributors in cancer research.
The manuscript appears to be scientifically, technically, and ethically valid. The authors utilized appropriate methods by conducting a systematic search of the literature and assessing studies for inclusion using well-defined criteria. Their analysis appears robust, with data extracted by multiple reviewers to minimize bias. There are no apparent ethical issues.
The review offers several merits:
· It provides a timely, comprehensive overview of active patient engagement in oncology trial design, an evolving area of research.
· The authors synthesize evidence across cancer types and stages, offering a broad perspective.
· They highlight tangible impacts of patient involvement, lending credibility to this approach.
· The proposed framework integrates patient insights across all trial stages, advancing conceptual thinking.
· Limitations and challenges are acknowledged, demonstrating scholarly rigor.
The manuscript fills a gap in the literature by consolidating the emerging research on patient involvement in trial design, whereas existing reviews focus on participation. It adds value by systematically demonstrating the benefits and feasibility of patient engagement. The originality of the manuscript stems from the focus on active involvement specifically in design stages, rather than participation alone. This advances the conceptualization of patients as collaborators. By emphasizing patient-centeredness and partnerships, the review could impact cancer research culture. It compellingly argues for maximizing patient contributions, though impact will depend on uptake by regulators and researchers.
Overall, this is a well-conducted review that makes a persuasive case for patient involvement throughout clinical trials. It synthesizes current evidence while advocating for more patient-centric research. With constructive revisions during peer review, it could make a meaningful contribution to the literature.
While the manuscript provides valuable insights, there are a few areas that could be refined to further augment the quality and impact of the work. Here are some respectful suggestions that could potentially improve the manuscript if the authors choose to implement them:
Introduction:
· In the Introduction, the authors may wish to expand upon recent trends and trajectories pertaining specifically to the integration of patient-centricity principles within oncology research contexts. Doing so would serve to provide enhanced contextual framing and rationalization for the focused examination of patient engagement in the design of oncology clinical trials.
· Within the Introduction, consideration could be given to incorporating brief coverage of any existing policies, regulations, or guidelines enacted by governance bodies, such as research councils or funders, regarding formalized integration of patient engagement in clinical trial designs and processes. Highlighting established protocols would aid readers in appreciating the manuscript's role in addressing this issue and emphasizing the need for further uptake of patient-engagement practices.
Methods:
· Further discourse on the rationale for selection of a scoping review methodology versus a systematic review would serve to strengthen the methodological justification. Articulating the basis for this methodological decision will enhance the scientific rigor underpinning the review.
· The authors may consider elaborating on any appraisals conducted to evaluate the qualitative robustness and risk of bias among the included studies. Discussing assessments of study quality amplifies the methodological rigor applied within the review.
Results:
· To further enhance the clarity of data synthesis, presenting descriptive quantitative summaries, such as quantification of studies categorized by cancer type or engagement methodology, may be beneficial. Incorporating numeric descriptions of the captured data can complement the qualitative findings.
Discussion:
· To further exhibit scholarly rigor, the authors could consider expanded discussion of limitations inherent both within the original evidence base itself as well as restrictions of the scoping review methodology. Articulating these limitations enables a balanced, critical appraisal.
· Building upon the noted challenges surrounding effective patient engagement, the authors have the opportunity to put forth potential strategies, approaches, or solutions to mitigate such obstacles. Proposing practical pathways to address engagement barriers would further enhance the applied value of the review.
Conclusions:
· To promote continuity, the authors could outline potential avenues for impactful future research endeavours to build upon the findings and framework presented. Elucidating logical next steps for investigation fosters advancement in this domain.
Thank you for allowing me to provide an academic critique of this unpublished manuscript. I am pleased to have had the chance to review such an interesting paper and hope that my feedback will support the authors in strengthening their work prior to potential publication.
Author Response
Introduction
Comment: In the Introduction, the authors may wish to expand upon recent trends and trajectories pertaining specifically to the integration of patient-centricity principles within oncology research contexts. Doing so would serve to provide enhanced contextual framing and rationalization for the focused examination of patient engagement in the design of oncology clinical trials.
Response: We are very thankful for this suggestion. In direct response, we have added a segment to our Introduction that outlines the recent advancements in patient-centric approaches within oncology research. We specifically highlighted the growing emphasis on personalized medicine and tailored therapies, helping to situate our work within contemporary shifts in the field [page 2, lines 59-63].
Comment: Within the Introduction, consideration could be given to incorporating brief coverage of any existing policies, regulations, or guidelines enacted by governance bodies, such as research councils or funders, regarding formalized integration of patient engagement in clinical trial designs and processes. Highlighting established protocols would aid readers in appreciating the manuscript's role in addressing this issue and emphasizing the need for further uptake of patient-engagement practices.
Response: We appreciate the reviewer’s input on this matter. We have expanded the Introduction to include specific guidelines and calls to action issued by major regulatory bodies such as CADTH, NIH, and EMA. We also included the contributions of agencies like PCORI, which have developed detailed methodologies for involving patients in clinical research. This new material underscores that active patient involvement is becoming a standard practice in oncology research, rather than an optional addition [page 2, lines 63-72].
Methods
Comment: Further discourse on the rationale for selection of a scoping review methodology versus a systematic review would serve to strengthen the methodological justification. Articulating the basis for this methodological decision will enhance the scientific rigor underpinning the review.
Response: We are very thankful for this suggestion. To address it, we added a section that explicitly outlines the reasons for choosing a scoping review methodology. Specifically, we emphasized that a scoping review allows for the inclusion of a diverse array of study designs and methodologies, offering a more comprehensive overview of the literature. This flexibility is particularly useful for our research question, which seeks to explore the multi-faceted and evolving nature of patient involvement in oncology clinical trials [page 2, lines 89-95].
Comment: The authors may consider elaborating on any appraisals conducted to evaluate the qualitative robustness and risk of bias among the included studies. Discussing assessments of study quality amplifies the methodological rigor applied within the review.
Response: We are very thankful for your guidance on enhancing the methodological rigor of our work. In response, we clarified in the manuscript that, being a scoping review, our primary aim was to provide an overview of the existing evidence, rather than to assess the quality of the individual studies included. This was articulated to ensure that readers understand the focus and limitations of a scoping review in this context.
Results
Comment: To further enhance the clarity of data synthesis, presenting descriptive quantitative summaries, such as quantification of studies categorized by cancer type or engagement methodology, may be beneficial. Incorporating numeric descriptions of the captured data can complement the qualitative findings.
Response: We are very thankful for this constructive suggestion. We concur that providing numeric descriptions can offer an additional layer of insight, aiding readers in easily grasping the scope and distribution of the existing literature. Accordingly, we have revised the Results section to include descriptive quantitative summaries. Now, the studies are enumerated and categorized based on cancer type and engagement methodology. These numeric descriptors serve to complement our qualitative findings, providing a more rounded understanding of the data [page 5-6, lines 156-229].
Discussion
Comment: To further exhibit scholarly rigor, the authors could consider expanded discussion of limitations inherent both within the original evidence base itself as well as restrictions of the scoping review methodology. Articulating these limitations enables a balanced, critical appraisal.
Response: We are very thankful for your guidance. Considering your suggestion, we expanded the discussion of the limitations inherent in our review. We acknowledged that focusing solely on published articles may exclude grey literature and that the scoping review methodology prevents robust risk-of-bias assessments. However, we also noted the strength of the scoping methodology in capturing a broader array of studies, thereby offering a more comprehensive view [page 9-10, lines 406-417].
Comment: Building upon the noted challenges surrounding effective patient engagement, the authors have the opportunity to put forth potential strategies, approaches, or solutions to mitigate such obstacles. Proposing practical pathways to address engagement barriers would further enhance the applied value of the review.
Response: We are very thankful for this thoughtful suggestion. We included a section detailing several strategies aimed at mitigating challenges related to patient engagement. This includes making it a standard practice to involve patients in trial design and HTA processes, establishing a "Patient Advisory Board," utilizing digital platforms for more accessible patient consultations, and instituting diversity quotas. Importantly, we stressed that the quality of patient involvement is equally important, advocating for a level of scientific literacy among participating patients [page 9, lines 368-282].
Conclusions
Comment: To promote continuity, the authors could outline potential avenues for impactful future research endeavours to build upon the findings and framework presented. Elucidating logical next steps for investigation fosters advancement in this domain.
Response: We are very thankful for this thoughtful recommendation. We wholeheartedly agree that elucidating logical next steps for investigation can further the discourse in a meaningful way. Accordingly, we have added a section in our conclusions to address this point explicitly. Specifically, we proposed that future research could focus on developing standardized guidelines for patient involvement in oncology clinical trials, which would emphasize contributions from educated and scientifically literate patients. We also suggested that investigating the role of patient education programs could be a pivotal next step. This would offer new avenues for reconciling expert and lay perspectives, which is one of the challenges identified in our review [page 10, lines 431-436].
Reviewer 2 Report
1. In addition to the diagrammatic representation of the stages of a clinical trial and the potential points of patient involvement at each stage. All of these should be written in detail and the authors should describe the stages, the categories, the challenges, and the recommendations in a clear and concise manner. These will enhance the readability and quality of this review.
2. The authors should enhance the images' quality.
3. The authors may also wish to discuss the challenges and solutions associated with involving critically ill cancer patients?
Minor editing of English language is needed
Author Response
Comment: In addition to the diagrammatic representation of the stages of a clinical trial and the potential points of patient involvement at each stage. All of these should be written in detail and the authors should describe the stages, the categories, the challenges, and the recommendations in a clear and concise manner. These will enhance the readability and quality of this review.
Response: We thank the reviewer for this recommendation. Based on the reviewer’s recommendation, we've expanded upon each stage of the trial process, from the "Initial Trial Design and Development" to "Future Research and Trial Replication." Within each stage, we've discussed the unique opportunities for patient involvement, the challenges that may arise, and recommendations to address them. This expanded narrative seeks to clarify how patient involvement can enrich each phase of a clinical trial, providing a more comprehensive view than before [page 16, lines 288-309].
Comment: The authors should enhance the images' quality.
Response: We are very thankful for the reviewer’s input; the images have been regenerated for better quality.
Comment: The authors may also wish to discuss the challenges and solutions associated with involving critically ill cancer patients?
Response: We thank the reviewer for raising a critical point. Due to word count constraints, we've expanded briefly in the discussion section to address the unique challenges and solutions involving critically ill cancer patients, particularly in the context of the Canadian regulatory system. We've discussed how patients can help reshape access to trials, suggest accelerated paths to treatments, and contribute to research reform, despite the often-complex regulatory landscape they navigate.